# Anaemia and Congestion in Heart Failure: Correlations and Prognostic Role

**DOI:** 10.3390/biomedicines11030972

**Published:** 2023-03-21

**Authors:** Pietro Scicchitano, Massimo Iacoviello, Antonio Massari, Micaela De Palo, Angela Potenza, Raffaella Landriscina, Silvia Abruzzese, Maria Tangorra, Piero Guida, Marco Matteo Ciccone, Pasquale Caldarola, Francesco Massari

**Affiliations:** 1Cardiology Section, Hospital “F. Perinei” Altamura, 70022 Altamura, Italy; 2Cardiology Unit, Department of Medical and Surgical Sciences, University of Foggia, 71122 Foggia, Italy; 3Korian Italia, 70021 Acquaviva delle Fonti, Italy; 4Cardiac Surgery Unit, Policlinic University Hospital, 70124 Bari, Italy; 5Ospedale Generale Regionale “F. Miulli”, 70021 Acquaviva delle Fonti, Italy; 6Cardiology Unit, Policlinic University Hospital, Piazza Giulio Cesare 11, 70124 Bari, Italy; marcomatteo.ciccone@uniba.it; 7Cardiology Section, Hospital “S. Paolo”, 70124 Bari, Italy

**Keywords:** heart failure, BIVA, BNP, anaemia, prognosis, congestion

## Abstract

The aim of this study was to evaluate the relationship between anaemia and biomarkers of central/peripheral congestion in heart failure (HF) and the impact on mortality. We retrospectively evaluated 434 acute/chronic HF (AHF/CHF) patients. Anaemia was defined as haemoglobin levels <12 g/dL (women) or <13 g/dL (men). The brain natriuretic peptide (BNP) and hydration index (HI) were measured. The endpoint of the study was all-cause mortality. Anaemia occurred in 59% of patients with AHF and in 35% with CHF (*p* < 0.001) and showed a significant correlation with the NYHA functional class and renal function. BNP and HI were significantly higher in patients with anaemia than in those without anaemia. Independent predictors of anaemia included BNP, estimated creatinine clearance (eCrCL), and HI. The all-cause mortality rate was 21%, which was significantly higher in patients with anaemia than in those without anaemia (30% vs. 14%, *p* < 0.001; hazard ratio: 2.6). At multivariate Cox regression analysis, BNP, eCrCL, and HI were independent predictors for mortality (Hazard ratios: 1.0002, 0.97, and 1.05, respectively), while anaemia was not. Anaemia correlates with HF status, functional class, renal function, BNP, and HI. Anaemia was not an independent predictor for mortality, acting as a disease severity marker in congestive patients rather than as a predictor of death.

## 1. Introduction

Anaemia—defined as haemoglobin (Hb) levels <12 g/dL in women and <13 g/dL in men—is a common finding in heart failure (HF) patients, thus conditioning management, comorbidities occurrence, and clinical outcomes [1]. The prevalence in HF ranges from 15 to 55% in chronic HF (CHF) outpatients and from 15 to 61% in hospitalised or pre-transplant HF patients [2].

The exact impact of anaemia on outcomes in patients with HF is still a matter of debate. Groenveld et al. [3] calculated a 46% increase in all-cause mortality risk in patients with CHF and anaemia. Nevertheless, higher serum creatinine levels reduced the impact of anaemia on mortality rates. Similar results were found in the recent analysis of Xia et al. [4], who found increased 1-year mortality rates in anaemic patients suffering from HF with reduced (HFrEF, 25% increase) or preserved (HFpEF, 38% increase) left ventricle ejection fraction, while they calculated a 22% increase in the risk for HF hospitalisation. However, Abebe et al. [5] questioned the prognostic impact of anaemia in severe heart failure patients as other confounding factors might impact on patients’ outcomes. Similarly, anaemia seemed not to impact on 1-year mortality risk in patients with acute HF (AHF) [6].

Despite the multifactorial origin of anaemia in HF [1,7], the influence of congestion on haemoglobin evaluation and the impact of this relationship in the general assessment and prognosis of these patients are not fully understood. Congestion is the mainstay of management and risk stratification in patients suffering from HF [8,9]. The evaluation of congestion is even more challenging due to the limitations of commonly used biomarkers, signs, and symptoms of HF, which are adopted in order to precisely identify the degree of fluid accumulation. For instance, signs and symptoms of peripheral fluid accumulation (“peripheral congestion”) are most frequently late markers of congestion: peripheral oedema appears when the interstitial fluid volume rises to at least 4–5 kg of body weight [10]. Furthermore, patients with HF might only show pulmonary/systemic fluid accumulation, without any peripheral manifestations (“central congestion”), which increases difficulties in the diagnosis and evaluation of congestion [8,9,11]. The introduction of biomarkers tried to implement the gap on diagnosis: natriuretic peptides (NPs), soluble suppressor of tumorigenicity (sST2), carbohydrate antigen 125 (CA125), and cluster of differentiation 146 (CD146) have been proposed as possible biomarkers of central congestion [9]. On the other hand, the use of bioimpendace vector analysis (BIVA) might provide further insights in the general assessment of peripheral congestion beyond the commonest signs whose reproducibility is questionable [12,13,14].

The literature provides data about the relationship between anaemia and biomarkers of congestion such as brain natriuretic peptide (BNP) or plasma volume [15,16,17,18,19].

Several hypotheses tried to explain the role of anaemia in HF. An absolute and/or relative deficiency of iron and/or erythropoietin, alterations in the red blood cells volume, haemodilution, and other mechanisms might impact the occurrence of anaemia in patients with HF [20].

The Reduction of Events with Darbepoetin Alfa in Heart Failure (RED-HF) Trial did not demonstrate any improvement in clinical outcomes in patients with HF and mild-to-moderate anaemia who were treated with darbepoetin alfa [21], thus enforcing the concept of different pathophysiologic mechanisms underlying the relationship between anaemia and congestion [22,23].

Nevertheless, there are no studies dealing with the evaluation of the absolute value of anaemia at admission—whatever the underlying physiopathologic mechanism—as a determinant of prognosis in patients with HF and the influence of congestion status on this relationship.

The aims of the present study were to evaluate the relationship between anaemia and biomarkers of peripheral (by means of the hydration index [HI] as assessed by bioimpendace vector analysis [BIVA]) and central (as assessed by means of BNP) congestion in patients with HF and to evaluate the prognostic impact of anaemia on the all-cause mortality of patients with HF and the influence of congestion on such a relationship.

## 2. Materials and Methods

### 2.1. Study Patients

This was a retrospective study which reviewed clinical data from patients suffering from acute (AHF) or CHF who were admitted at our Department between January 2010 and November 2013. Specifically, we collected data from patients admitted to the ward for AHF and those who attended the CHF outpatient clinic which is associated with our Department.

The dataset gathered information about patients’ clinical characteristics, blood chemistry data, BIVA, and pharmacological treatments at baseline. The left ventricular ejection fraction (LVEF) was calculated by echocardiography (Simpson’s method). BNP levels were assessed using a microparticle enzyme immunoassay (Architect, Abbott Park, IL, USA). Serum creatinine was measured with a Beckman Coulter AU 680 chemistry analyser. All of these measurements were performed as a routine evaluation of the patients.

The exclusion criteria were: myocarditis, pericarditis, pulmonary embolism, acute coronary syndrome, recent cardiac surgery intervention, haemodialysis, and the use of erythrocyte stimulating agents or intravenous iron administration.

The primary endpoint was all-cause mortality, which was assessed from available medical records or National Death Records.

The study complied with the Declaration of Helsinki and was approved by the local Institutional Review Board. Written informed consent was obtained from each patient at inclusion (protocol n. 0081801/CE—29/10/2015, study number: 4816).

### 2.2. Creatinine-Based Estimated Glomerular Filtration Rate (eGFR)

We estimated the glomerular filtration rate (eGFR) by means of the Cockroft–Gault formula (mL/min/1.73 m^2^) as: (140—age in years) × (weight in Kg)/(72 × serum creatinine in mg/dL) × 0.85 (if female) [24] and, according to the clinical guidelines of the National Kidney Foundation, categorised the patients into four groups GFR: <30 mL/min/1.73 m^2^, 30–59 mL/min/1.73 m^2^, 60–90 mL/min/1.73 m^2^, and >90 mL/min/1.73 m^2^.

Although the gold standard for the evaluation of kidney function is based on the use of exogenous markers such as iothalamate or inulin, as these compounds are completely filtered by glomeruli with no secretion/absorption, their daily clinical application is far from being feasible [25]. Equations for the estimating glomerular filtration rate in heart failure seem to not be perfectly accurate, as their accuracy is lower than 75% [26]. Nevertheless, it is easy to include them in daily clinical practice, while the literature provided evidence for the prediction of adverse outcomes [27]. Szummer et al. [28] even demonstrated better predictive values of the Cockroft–Gault formula in predicting all-cause death in patients with HF.

### 2.3. Hydration Status Assessed by Bioimpedance Vector Analysis

Tetrapolar BIVA was assessed on the right side of the body by means of plethysmography emitting 50 kHz alternating sinusoidal current (CardioEFG, Akern RJL Systems, Florence, Italy), as previously reported [12,29]. The two vector components R and Xc of BIVA were divided by the subject’s height for the construction of an R/Xc graph that identified a vector as a result of interaction between two parameters. We estimated the hydration status by comparing the individual vector with sex-specific tolerance ellipses. Patients were considered “dry” if the vector fell into the 50th percentile vector tolerance or the upper pole of the ellipse and “wet” if it fell at the lower pole of the 50th percentile. In particular, it was “severe” when it was out of the lower pole of the 95th percentile, “moderate” when it was between the 95th and 75th percentiles, and “middle” when it was between the 75th and 50th percentiles. Additionally, using the equation that used two components, R and Xc (Bodygram 1.4, Akern RJL Systems, Florence, Italy), the device provided a quantitative hydration percentage (hydration index, HI%).

### 2.4. Statistical Analysis

Normally distributed variables were expressed as the mean (standard deviation), and non-normally distributed continuous variables were expressed as the median (95% confidence intervals (95% CI)). We log-transformed BNP values because they were not normally distributed (Shapiro–Wilk tests). Categorical variables were presented as the percentage (%). Comparisons between the groups were performed with Student’s *t*-test, a chi-square test, or one way analysis of variance (ANOVA), as appropriate. Pearson’s coefficient was used to evaluate the correlation between Hb levels and other variables. We performed univariate regression analysis in order to evaluate conditions that might be associated with the anaemic status of our patients. Those variables that showed a statistically significant association with anaemic status were computed into a multivariate analysis in order to identify independent predictors of anaemia in our HF population.

Odds ratios (OR) with 95% CI values were given. Receiver-operating characteristic (ROC) curve analysis was performed to calculate the area under the curve (AUC) values, while the optimal cut-off values for mortality were calculated by means of the Youden Index. Kaplan–Meier cumulative survival plots with a log-rank significance test was also performed.

Univariate and multivariate Cox proportional hazards regression models with estimations of hazard ratios (HR) and 95% CI were performed to evaluate the impact of variables on mortality. *P*-values below 0.05 were defined as statistically significant. Statistical analyses were performed using STATA software, version 12 (StataCorp, College Station, TX, USA).

## 3. Results

We enrolled 434 patients: 252 with CHF and 182 with AHF. Table 1 summarises the main characteristics of the study population.

Appendix A tries to show the differences in terms of the hydration index, BNP concentrations, and estimated creatinine clearance among all patients, those suffering with AHF, and those with CHF.

The patients with AHF showed significantly lower haemoglobin levels (13.2 ± 2 vs. 12 ± 2 g/dL, *p* < 0.001). The AHF patients demonstrated a higher prevalence of anaemia as compared to the CHF ones (Figure 1A). Specifically, the higher the NYHA functional class (i.e., the worse the HF condition of the patient), the higher the prevalence of anaemia (Figure 1B).

The relationship between renal function and anaemia occurrence in HF has been explored. Figure 2A,B outlines a higher prevalence in anaemic patients with eGFR deterioration (r = 0.41; *p* < 0.001).

Similar results were seen according to BNP plasma levels: BNP levels were significantly higher in patients with anaemia than in those without it (Figure 2C) and inversely related to haemoglobin concentrations (r = −0.36; *p* < 0.001) (Figure 2D). The evaluation of hydration status by means of HI revealed higher values in patients with anaemia (Figure 2E); specifically, HI was inversely related to haemoglobin levels (Figure 2F).

Univariate regression analysis was performed by involving all the anthropometric, clinical, laboratory, instrumental, and pharmacological characteristics of the study population in order to evaluate the possible association with anaemia. Those variables that were statistically significantly associated with anaemia at univariate regression analysis entered the multivariate analysis.

At multivariate logistic regression analysis, renal function, BNP, and hydration index remained independently associated with anaemic condition, while NYHA class and type of HF were not confirmed as determinants of anaemia (Table 2).

No further conditions were significantly associated with the prognosis of our patients. Specifically, diuretics did not show any relationship with congestion status nor with anaemia.

Indeed, a higher HI was associated with a 9% increase in the risk of anaemia, just as LnBNP and eGFR were associated with 43% and 25% increases in anaemic status, respectively. Higher values in the Wald test indicated the goodness of the statistically significant association of these three parameters with anaemic status.

Fifty patients died after a median follow-up of 463 days (IQR: 287–669). The cumulative mortality rate was 22% and occurred in 30% of patients with anaemia and 14% of patients without it (*p* < 0.001). At univariate Cox regression analysis, type of HF (acute vs. chronic), age, NYHA class, LVEF, renal function, anaemic condition, haemoglobin levels, BNP, and hydration status were predictors of mortality (Table 3).

The analysis of ROC curves identified haemoglobin ≤11.9 g/dL (AUC = 0.66, sensitivity 61%, and specificity 72%, *p* < 0.001) as the cut-off for the discrimination of mortality risk. Kaplan–Meier analysis indicated a significant increase in all-cause mortality in the anaemic group patients (Figure 3).

Nevertheless, NHYA class, AHF status, and anaemia declined their prognostic role at multivariate Cox regression analysis, while BNP, eCrCL and HI confirmed their independent predictive value for mortality (Table 3).

## 4. Discussion

Anaemia is a crucial finding in patients suffering from HF [1,30,31]. Despite many studies correlating anaemia with the prognosis of patients with HF [31,32,33,34], there is a paucity of data related to the interplay between anaemia and congestion, i.e., the most important clinical feature of patients with HF [15,16,17,18].

This is the first study that tried to evaluate the relationship between anaemia and peripheral/central congestion in patients with AHF or CHF. The main findings from our analysis were the following: 1. anaemia was a common finding in HF patients, and its prevalence was higher in patients in their acute phase of HF; 2. anaemia was directly related to peripheral (as assessed by BIVA) and central (as indicated by BNP plasma concentrations) congestion in HF; and 3. at multivariate regression analysis, anaemia did not impact prognosis when congestion parameters were included in the analysis.

We adopted BNP as a biomarker of central congestion in relation to laboratory issues: BNP is routinely performed in our Department/Outpatient Unit in patients with HF as clinical practice, while no further serum biochemical biomarkers have been considered as substitutes or in association with BNP. Indeed, BIVA evaluation is part of the clinical assessment of peripheral congestion in all patients who were admitted to our Department/Outpatient Unit.

Our study pointed out a percentage in anaemia equal to 45% in the whole population, although the rate was significantly higher in AHF (59%) than in CHF patients (35%, *p* < 0.001). This was confirmed by the analysis of anaemia prevalence in relation to NYHA class: the higher the NYHA class, the higher the prevalence in anaemia (Figure 1). When considering studies that adopted the WHO definition for anaemia, its occurrence ranged from 13.5% to 45% in patients with CHF and from 20.6% to 46% in AHF [2,35].

The Italian IN-CHF registry demonstrated a prevalence of anaemia in HF equal to 15.5%, while the Valsartan Heart Failure Trial (Val-HeFT) showed a prevalence equal to 9.9% [33]. Similar results were reported by Tanner et al. [36], who observed a percentage equal to 15% in the prevalence of anaemia in HF. Finally, an Indian registry found an anaemia prevalence of 35.8% in their HF population [37].

The idea that haemodilution could be the cause of this impairment has been investigated. Abramov et al. [15] demonstrated no reduction in haemoglobin levels in patients with HF but rather the occurrence of relative reduction due to the increase in plasma volume in patients with HFrEF. The occurrence of cardio-renal syndrome might also explain anaemia in HF and its impact on prognosis [38]: chronic kidney disease (CKD) is related to the occurrence of anaemia, and kidney hypoperfusion might directly induce anaemia. A reduction in erythropoietin production might partially explain such mechanisms despite the RED-HF trial [21] not completely supporting this hypothesis in terms of the prognosis and impact on outcomes. Westenbrink et al. [22] identified fluid retention as a determinant for anaemia in HF, in association with impaired renal perfusion and a reduction in erythropoietin production. Adlbrecht et al. [23] demonstrated that fluid overload in patients with HF acted as an independent risk factor for the occurrence of anaemia.

Anaemia in HF might also exacerbate the decompensation of HF: chronic anaemia has been linked to the retention of salt and water, a reduction in renal blood flow and the glomerular filtration rate, and neurohormonal activation, which implement the congestion status commonly observed in HF patients [17]. Krzesiński et al. [39] effectively demonstrated increased thoracic fluid content—as assessed by impedance cardiography—in patients with AHF, as well as higher NT-proBNP. Our data were in line with these findings: we observed higher HI values—as assessed by BIVA (Figure 2E,F)—in HF patients with anaemia, as well as increased values of BNP (Figure 2C,D).

Desai et al. [16] found an inverse relationship between haemoglobin and NT-proBNP that was independent of possible confounding factors such as cardiovascular risk factors, ventricular function, myocardial ischaemia, inflammation, and kidney function. Similar results were found in the analysis of Wu et al. [18]: there was an inverse correlation between log BNP and haemoglobin in patients with diastolic HF, while anaemic status was directly related to the worst NYHA classes. Indeed, we further considered BIVA evaluation for the assessment of peripheral congestion. We already demonstrated the accuracy of BIVA in detecting peripheral congestion better than oedema identification [12,29]. Piccoli [13] outlined the possibility of detecting, monitoring, and controlling the congestion burden of patients with HF and/or severe kidney impairment by analysing the vector displacement. González-Islas [40] demonstrated the impact of BIVA on evaluating differences in body composition in HF, thus observing the possibility of using bioimpedance analysis for the assessment of congestion in HF. Finally, BIVA might provide prognostic information for patients with HF, above all, when combined with further biomarkers [41].

Hydration index [HI] might effectively be considered as an early instrument for the detection of peripheral congestion in HF as compared to the commonest signs and symptoms of peripheral congestion [12,14,29,42]. Peripheral congestion is characterised by fluid accumulation in peripheral districts. Nevertheless, physiopathological considerations outline the need for 3 to 5 L of fluid accumulation before peripheral oedema occurs [10]. The diagnostic accuracy of oedema—the most notably marker of peripheral congestion—is poor: sensitivity 46%, specificity 73% [8]. Nevertheless, we previously demonstrated that HI showed 79% and 82% in sensitivity and specificity, respectively, in AHF and 85% and 80% in CHF [12]. Therefore, we do consider HI as a fundamental tool in the diagnosis and management of patients with overt and—above all—subtle congestive HF.

Similar considerations are according to biomarkers of central congestion. Clinical signs and symptoms of central congestion (i.e., dyspnoea on exertion, orthopnoea, resting jugular vein distension, S3) did not provide sufficient data on their accuracy [8]. They demonstrated higher specificity or sensitivity but not both of them, thus reducing their effective impact on clinical decision making [8]. On the other hand, biomarkers provided better insights. BNP demonstrated 95% sensitivity and 67% specificity when a lower threshold was considered for HF diagnosis [43]. NT-pro BNP seemed more effective than sST2 in HF diagnosis: the BNP area-under-the-curve (AUC) for diagnosing HF with preserved ejection fraction was higher than the sST2 AUC (0.881 vs. 0.717), as well as in the case of patients with HF with reduced ejection fraction (0.945 vs. 0.820) [44].

Indeed, fluid retention in HF is a complex physiopathologic condition with many actors [45]. The renin–angiotensin–aldosterone system (RAAS) principally impacts systemic fluid regulation: it is responsible for tubular sodium reabsorption and concomitant water retention; the action of angiotensin II on angiotensin receptor 1 (AT1) induces glomerulus hypoperfusion due to the vasoconstriction of the afferent and efferent arterioles: this reduces the filtration rate of the kidney and promotes systemic fluid accumulation [46]. Indeed, hypoperfusion of the kidney due to RAAS activation may lead to an increase in oxygen demand and thereby stimulate erythropoietin (EPO) production [47]. Therefore, medications for HF might promote anaemia occurrence in HF. Furthermore, angiotensin II is able to promote the release of the antidiuretic hormone (ADH): the role of this hormone is to induce the retention of water by the collecting duct and increase extracellular fluid [46].

The NP system is also able to impact both fluid retention and anaemia in patients with HF [45]. HF increases the production of both atrial (ANP) and BNP, which in turn activate natriuretic peptide receptor B (NPR-B). This interaction promotes natriuresis and diuresis, thus dramatically impacting the cardiac preload [48]. It has been established that, in healthy individuals, the occurrence of anaemia is related to increased BNP plasma values [49]. Similar results were found in patients with HF: an inverse relationship was observed between NT-pro BNP values and haemoglobin concentration [50]. The combination of increased BNP plasma values and anaemia significantly increased the risk for major adverse cardiac events [51].

These relationships might be related to fluid retention, but evidence demonstrated that ANP/BNP promoted the inhibition of the RAAS, thus reducing EPO production [52]. Similar considerations are in relation to the inhibition of the sympathetic tone promoted by increased plasma levels in ANP/BNP in patients with HF [52].

We observed a relationship between peripheral congestion and anaemia in patients with HF. It is interesting to observe that anaemia did not confirm its predictive value for all-cause mortality at multivariate Cox regression analysis when the glomerular filtration rate and peripheral and central congestion biomarkers were added to the final model.

This final result is interesting: anaemia lost its predictive value as compared to congestion biomarkers. We can suppose that—no matter the haemoglobin levels in HF—physicians should pay more attention to the congestion evaluation than the anaemia status for impacting all-cause mortality in patients with AHF and CHF. Naturally, this is a hypothesis-generating overview on the basis of this retrospective analysis. Despite several studies dealing with the impact of anaemia on outcomes of patients with HF [53,54], the lack of a correlation with congestion status is worrisome. Further randomised controlled trials are needed to evaluate the role of congestion and/or anaemia in HF patients, with the purpose of delineating the real determinants of their prognosis as well as better focusing the attention of physicians to rightly correct abnormalities in patients with HF.

## 5. Limitations

This manuscript has some limitations. The retrospective nature of the paper is a limitation, although this design allowed us to collect data from a wide number of patients with AHF and CHF. Furthermore, this was a single-centre study: including more centres would increase the sample size and the robustness of the data.

## 6. Conclusions

Anaemia is a hallmark in patients with HF. Its prevalence is higher in the acute setting, and it is related to kidney function and biomarkers of central and peripheral congestion. Indeed, congestion and kidney function confirmed their predictive role in the all-cause mortality rate in patients with HF, rather than the anaemic status.

## Figures and Tables

**Figure 1 biomedicines-11-00972-f001:**
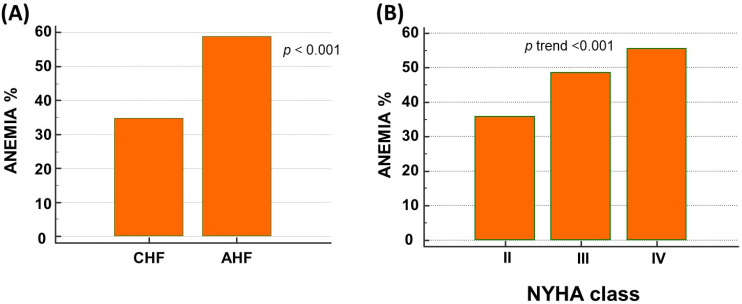
(**A**) Prevalence of anaemia in patients with acute (AHF) and chronic heart failure (CHF). (**B**) Relationship between anaemia and NYHA functional class.

**Figure 2 biomedicines-11-00972-f002:**
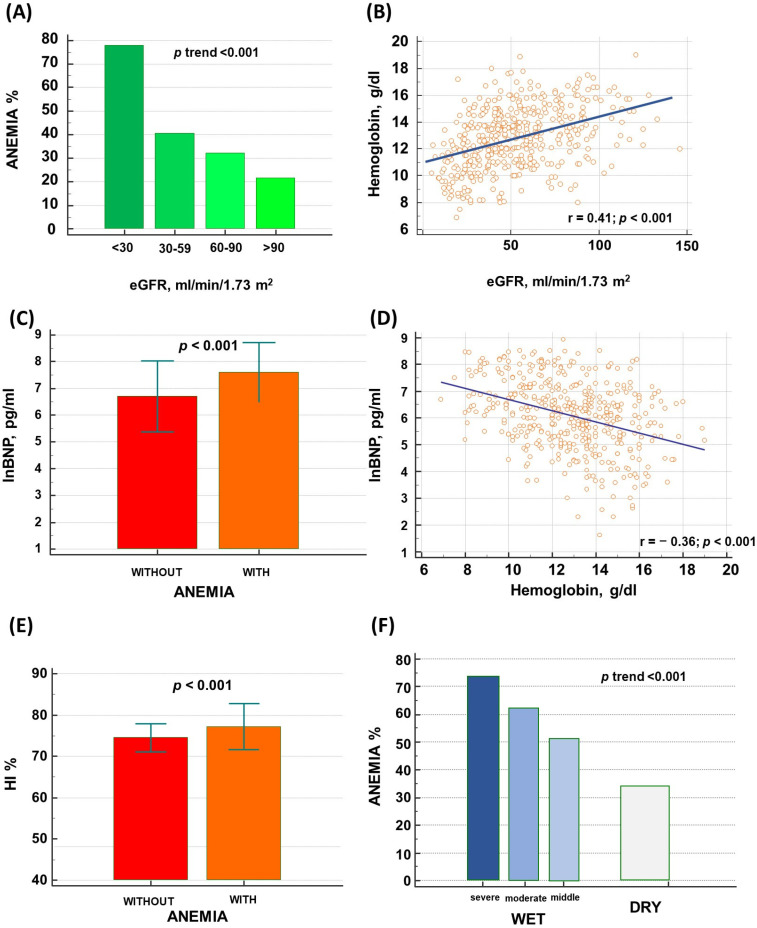
(**A**) Relationship between anaemia and renal function, the latter categorised into four groups according to the values of the estimated glomerular filtration rate. (**B**) Relationship between renal function—considered as a continuous variable—and hemoglobin levels. Comparison of BNP concentration between patients with and without anaemia (**C**), and correlation between BNP and hemoglobin levels (**D**). Comparison of hydration status between patients with and without anaemia (**E**), and correlation between hydration status and hemoglobin levels (**F**).

**Figure 3 biomedicines-11-00972-f003:**
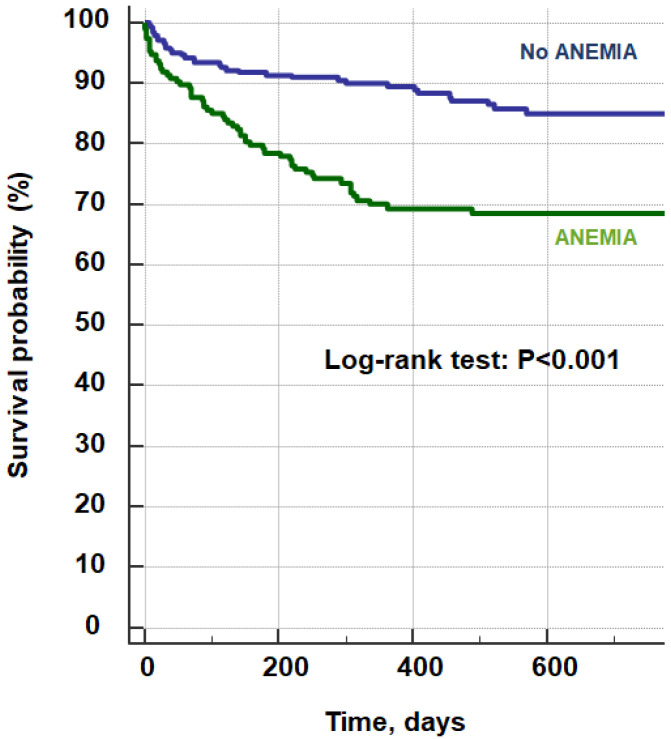
Kaplan–Meier survival curves stratified for the absence or presence of anaemia.

**Table 1 biomedicines-11-00972-t001:** Patient characteristics of all patients.

Clinical Characteristics	Overall (n = 434)
Age, yrs	75 ± 11
Male/Female, %	52/48
BMI, kg/mq	28 ± 5
NYHA class II/III/IV, %	43/30/27
Peripheral edema, %	30

Medical history, %	
Coronary artery disease	31
Diabetes	24
Atrial fibrillation	43
CKD	33
COPD	21
AHF	42
ICD	11
Instrumental Evaluations	
LVEF	42 ± 12%
Preserved LVEF, %	48
Mid-range LVEF, %	10
Reduced LVEF, %	42
BIVA, hydration index, %	76 ± 5
Laboratory values	
BNP, pg/mL, median (CI)	516 (423–582)
Hemoglobin, g/dL	13 ± 2
Anaemia, %	45
BUN, mg/dL	30 ± 17
Uric acid, mg/dL	6.2 ± 2
Creatinine, mg/dL	1.4 ± 0.8
Creatinine > 1.5 mg/dL, %	20
eCrCl, mL/min per 1.73 m^2^	57 ± 29
eCrCl, <60 mL/min per 1.73 m^2^, %	59
eCrCl, <30 mL/min per 1.73 m^2^,%	18
Sodium, mmol/L	139 ± 4
Potassium, mmol/L	4.0 ± 0.6
Therapies, %	
Furosemide	70
Beta-blockers	50
ACE inhibitors	39
ARBs	21
MRAs	69
Digitalis	21
Ivabradine	5
IV inotropes	5

Abbreviations: ACE: angiotensin-converting enzyme; AHF: acute heart failure; ARB: angiotensin receptor blocker; BMI: body mass index; BUN: blood urea nitrogen; CKD: chronic kidney disease; COPD: chronic obstructive pulmonary disease; eCrCl: estimate creatinine clearance; ICD: implanted cardioverter/defibrillator; IV: intravenous; LVEF: left ventricular ejection fraction; MRAs: mineralocorticoid receptor antagonists; NYHA: New York Heart Association.

**Table 2 biomedicines-11-00972-t002:** Predictors of anemic status in logistic multivariate regression analysis.

Variables	Odds Ratio (95% CI)	*p*	*B* Coefficient	SE	Wald
AHF vs. CHF	1.25 (0.74–2.11)	0.4			
NYHA class	0.83 (0.62–1.10)	0.2			
Hydration index, %	1.09 (1.03–1.15)	=0.0008	0.09	0.03	11.2
LnBNP, pg/mL	1.43 (1.14–1.79)	=0.002	0.36	0.11	9.7
eGFR, mL/min	1.25 (1.01–1.44)	=0.001	0.22	0.071	10.3

Abbreviations: AHF: acute heart failure; LnBNP: natural logarithmic brain natriuretic peptide; CHF: chronic heart failure; CI: confidential interval; eGFR: estimate glomerular filtration rate; NYHA: New York Heart Association; SE: standard error; Wald: Wald test.

**Table 3 biomedicines-11-00972-t003:** Univariate and multivariate Cox proportional hazards survival analyses.

	Univariate Cox Regression Analysis		Adjusted Cox Regression Analysis		
	HR (95% CI)	*p*	HR (95% CI)	*p*	Wald
AHF vs. CHF	2.70 (1.77–4.14)	<0.0001			
Age, year	1.07 (1.05–1.10)	<0.0001			
NYHA class	1.80 (1.40–2.30)	<0.0001			
LVEF, %	0.99 (0.97–1.01)	=0.2			
Anaemia, yes vs. no	2.55 (1.66–3.91)	<0.0001			
Hemoglobin, g/dL	0.78 (0.70–0.85)	<0.0001			
Hydration Index, %	1.11 (1.07–1.15)	<0.0001	1.05 (1.005–1.08)	=0.03	4.8
BNP, pg/mL	1.0004 (1.0003–1.0005)	<0.0001	1.0002 (1.0001–1.0004)	=0.1	6.4
eGFR, mL/min	0.96 (0.95–0.97)	<0.0001	0.97 (0.96–0.99)	=0.0006	11.7

Abbreviations: AHF: acute heart failure; BNP: brain natriuretic peptide; CHF: chronic heart failure; CI: confidential interval; eGFR: estimate glomerular filtration rate; HR: hazard ratio; LVEF: left ventricular ejection fraction; NYHA: New York Heart Association.

## Data Availability

Data will be available on request by contacting the corresponding author.

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
