# Peer review of "Anaemia and Congestion in Heart Failure: Correlations and Prognostic Role"

_biomedicines, 2023, doi:10.3390/biomedicines11030972_

Round 1
Reviewer 1 Report
This manuscript provides an interesting issue regarding the relation of anaemia and congestion in the stages of heart faliure as possibly prognostic values in a retrospective study.
Main issues
Please provide more information on the definition of „central and peripheral congestion markers” as mentioned in the text. Also please explain why BNP alone is attributed to „central marker” related to congestion and HF. Related to this issue, the following topics are suggested to discuss further in the text:
1. Can the detected anaemia be the consequence of congestion in higher stages of HF?
2. What was the relationship of hydration/congestion and diuretic therapies in HF?
3. eGFR is mentioned in relation to kidney function. Can you provide more information on the diagnostic tools evaluating kidney function? Can eGFR be alone a full diagnostic indicator?
Based on the arised issues, a further extension of Discussion would be suggested in the following topics:
1. Since congestion and hydration is addressed, it is suggested to discuss other hormones and parameters to provide more information in this topic. As suggested above, more details in the control of fluid homeostasis in relation with other hormones like ADH, RAS (with angiotensin II), ANP would be discussed and also their alterations in congestion and HF.
2. Please give more explanation on the issue of „central and peripheral congestion markers”.
3. a Summary Figure is suggested to provide a better understanding of the Conclusions.
Please check the whole text extensively for English.
Please improve the quality of the Figures.
Specific issues:
Abstract. Please edit HF. Please replace HR to other abbrev, e.g. Hr, as HR indicates „heart rate” most of all.
line 28 please check for English grammar. Please check sentences to make logical connections in the text. e.g. lines 23-24. „We evaluated…” Why?, „The endpoint…” of what?, etc.
Lines 62-66: The definition of „biomarkers of peripheral and central congestion” is a bit far since 1-1 markers have been examined in this respect. It would be better to provide also the relevant specific markers in this issue. However this issue needs further explanation as suggested above.
Lines from 115 and related Results: Please provide more information regarding the outcome of this regression analysis and please explain more details in this respect of Table 2 in Results.
Table 1, please give subsections of subtopics (Med. History, Lab values, Therapies) in the table.
Fig 1. Please explain also criterias of AHF, CHF, stages of NYHA in HF in the clinical point of view. Please provide a better quality.
FIg. 2. Please provide a better quality. Lines 150-151 please check for English grammar.
Table 2. Please provide more explanation here about the regression results (e.g. how each is associated with anaemic condition). Please edit Wald in Legends.
Fig 3. Please adjust Fig format to others. (e.g. less strong letters)
Please correct reference list (omit numbering of the last chapter). Please provide more references in the topics arised (e.g. hormonal control of fluid homeostasis and its changes in HF, more clinical point of view of HF and its treatment, indicator parameters of congestion, etc.).
Author Response
Reviewer #1
We thank this Reviewer for the constructive comments and suggestions. Furthermore, we would like to really thank him/her for his/her appreciation about our research. This is our point to point reply.
- Please provide more information on the definition of „central and peripheral congestion markers” as mentioned in the text. Also please explain why BNP alone is attributed to „central marker” related to congestion and HF. Related to this issue, the following topics are suggested to discuss further in the text:
- Can the detected anaemia be the consequence of congestion in higher stages of HF?
Thank you very much for this useful and interesting insights which certainly improve the entire discussion about this hot topic in HF. We do believe that anaemia might be consequence of the congestion status above all in higher stages of HF. Literature provides studies about this relationship and we agree with the pathophysiological background at the base of this concept. We tried to implement the introduction and discussion section in order to better include a brief overview about the influence of congestion on the occurrence of anaemia in patients with HF. Indeed, the real aim of our paper was to better consider and discuss the absolute value of anaemic identification in patients with HF on prognosis and identify a possible linkage with congestion.
- What was the relationship of hydration/congestion and diuretic therapies in HF?
That’s a good insight and we really thank the reviewer for this comment. Indeed, we did not find any relationship between hydration/congestion status and diuretic therapies in our analysis. We included a specific statement in the results section in order to outline this point.
- eGFR is mentioned in relation to kidney function. Can you provide more information on the diagnostic tools evaluating kidney function? Can eGFR be alone a full diagnostic indicator?
Thanks for this comment. eGFR alone could effectively not be a full diagnostic indicator of renal function. Nevertheless, eGFR is the most used indicator in daily clinical practice. Due to evidence about the prognostic role of eGFR, it is widely accepted as indicator of renal function. Therefore, we conventionally used eGFR as diagnostic tool evaluation kidney function. We discussed such a point in the methods section.
- Based on the arised issues, a further extension of Discussion would be suggested in the following topics:
- Since congestion and hydration is addressed, it is suggested to discuss other hormones and parameters to provide more information in this topic. As suggested above, more details in the control of fluid homeostasis in relation with other hormones like ADH, RAS (with angiotensin II), ANP would be discussed and also their alterations in congestion and HF.
We updated the discussion section by providing evidence about the relationship among anemia, congestion, and fluid homeostasis controllers.
- Please give more explanation on the issue of „central and peripheral congestion markers”.
Thanks for the suggestion. We effectively implemented introduction and discussion section in order to better explain “central and peripheral congestion” markers in HF. Thank you for this comment.
- a Summary Figure is suggested to provide a better understanding of the Conclusions.
Thanks for your suggestion. We completely revised the graphical abstract in order to better summarize the conclusions of the paper.
- Please check the whole text extensively for English.
Thank you for the suggestion. We further revised the English of the text.
- Please improve the quality of the Figures.
Thank you for the comment. We tried to ameliorate the quality of the figures.
Specific issues:
- Please edit HF. Please replace HR to other abbrev, e.g. Hr, as HR indicates „heart rate” most of all.
Thank you for the suggestion. We revised the abstract section as per your indications.
- line 28 please check for English grammar. Please check sentences to make logical connections in the text. e.g. lines 23-24. „We evaluated…” Why?, „The endpoint…” of what?, etc.
Thanks for the indications. We amend the typos.
- Lines 62-66: The definition of „biomarkers of peripheral and central congestion” is a bit far since 1-1 markers have been examined in this respect. It would be better to provide also the relevant specific markers in this issue. However this issue needs further explanation as suggested above.
Thank you once again for this remark. We improved this paragraph in order to improve the comprehension of the text and its content.
- Lines from 115 and related Results: Please provide more information regarding the outcome of this regression analysis and please explain more details in this respect of Table 2 in Results.
Thank you for this comment. We implemented this section in order to improve its readability.
- Table 1, please give subsections of subtopics (Med. History, Lab values, Therapies) in the table.
We maintain the empty spaces in order to underline the subsection of the subtopics.
- Fig 1. Please explain also criterias of AHF, CHF, stages of NYHA in HF in the clinical point of view. Please provide a better quality.
We included the criteria for AHF and CHF in the methods section when we described the enrolled population. We also include a brief sentence for describing the relationship among NYHA class, anaemia, and HF.
- 2. Please provide a better quality. Lines 150-151 please check for English grammar.
Thanks once again for this comment. We revised the figure and the sentences.
- Table 2. Please provide more explanation here about the regression results (e.g. how each is associated with anaemic condition). Please edit Wald in Legends.
We improved this section in order to better describe the results.
- Fig 3. Please adjust Fig format to others. (e.g. less strong letters)
Thank you very much. We amend the figure.
- Please correct reference list (omit numbering of the last chapter). Please provide more references in the topics arised (e.g. hormonal control of fluid homeostasis and its changes in HF, more clinical point of view of HF and its treatment, indicator parameters of congestion, etc.).
Thank you for this comment. The reference list has been formatted in agreement with the Journal standards and submission guidelines. References not included in the text had been deleted. The list has been extended in order to include more references related to the topics pointed out by the reviewer.
Reviewer 2 Report
The present study “Anaemia and Congestion in Heart Failure: Correlations and Prognostic Role” is interesting however the authors should address the following concerns.
· Authors are suggested to include female % in Table 1
· Authors should also mention the type of medication these subjects were using and if there are any exclusion or inclusion criteria
Author Response
Reviewer #2
We thank this Reviewer for her/his useful suggestions. We sincerely appreciate his/her comments on our work. This is our point-to-point reply:
- Authors are suggested to include female % in Table 1
Thanks for the suggestion. We updated table 1.
- Authors should also mention the type of medication these subjects were using and if there are any exclusion or inclusion criteria
Thanks for the comment. We included the type of medication in table 1. Exclusion criteria had been included in the Methods section.
Round 2
Reviewer 1 Report
The paper has been extensively improved. Just minor spell check is required.